# Online Mathematics Education during the COVID-19 Pandemic: Didactic Strategies, Educational Resources, and Educational Contexts

**Ronnie Videla** [1,2,3,*], **Sebastián Rossel** [1,2,3], **Coralina Muñoz** [3,4] **and Claudio Aguayo** [5]

1   Facultad de Educación, Escuela de Educación Diferencial, Universidad Santo Tomás,
    La Serena 1780000, Chile; srossel@userena.cl
2   Facultad de Humanidades, Departemento de Educación, Universidad de La Serena, La Serena 1780000, Chile
3   INNOVA STEAM Lab, La Serena 1780000, Chile; coralina.mvidal@gmail.com
4   Centro de Experimentación, Perfeccionamiento e Investigaciones Pedagógicas CPEIP, Santiago 8320000, Chile
5   AppLab, Te Ara Poutama, Faculty of Māori and Indigenous Development,
    Auckland University of Technology, Auckland 1010, New Zealand; claudio.aguayo@aut.ac.nz
*   Correspondence: rvidela@userena.cl

**Abstract:** One of the impacts of the COVID-19 pandemic has been restrictions on mobility and thus the closure of schools. This has had consequences on the teaching strategies of primary mathematics educators who were not familiar with online education. Most schools in Chile have adopted virtual and hybrid classes to continue educational processes. From a quantitative approach with a sample of $n$ = 105 primary school educators and through an online survey, we analyzed how educators implemented the mathematics curriculum during the pandemic using various didactic strategies and educational resources, as well as their respective contexts. The results show that there is a relationship between the level of technical knowledge of teachers, the years of experience, and the types of teaching strategies they use. Likewise, differences were found between educators in rural and urban sectors according to the use of teaching strategies and the types of educational resources used. Regarding the didactic strategies, it is shown that the emerging strategies most used are metaphorical and analogical, whereas in traditional strategies the automation of procedures is imposed. The implications for practice include suggestions and guidelines for improving the training and professional development of mathematics teachers including increasing and strengthening the number and quality of teachers' didactic strategies and online pedagogical management skills and promoting metacognition through virtual forums. Finally, we discuss the context of the use of didactic strategies in mathematics during the pandemic, analyzing its challenges and opportunities.

**Keywords:** mathematics education; online teaching; teaching strategies; primary education; COVID-19

## 1. Introduction

One of the impacts of the COVID-19 pandemic has been restrictions on mobility and thus the closure of schools [1]. This has led to education being carried out through different types of virtual education platforms from a distance using pedagogical support from families, especially at preschool and primary education levels [2]. The impact of the pandemic on the transformation of distance education processes compared to face-to-face teaching not only translates into sudden changes and the adaptation of new virtual teaching and learning methods but also generates mental health problems for students. In addition, primary school students show reluctance and little concern for homework [3]. This becomes another demand for educators who must deal not only with changes in their teaching but also in many cases also be involved with the educational processes of their own families [4]. Mathematics education has been one of the disciplines that has been most in need of the use of new didactic teaching strategies considering its symbolic nature and given its emphasis on paper rather than technology [5].

A study carried out by The Royal Society of Mathematics in London (2020) on the impact of the pandemic on mathematics education in England, revealed low motivation and anxiety in students toward tasks as well as difficulties faced by teachers in remotely monitoring learning progress, which had a long-term negative impact on student learning. The same study reported that two out of three math students are between 1 and 6 months behind in their learning in what could have been a face-to-face teaching process. The panorama is even more discouraging when knowing the UNESCO figures that estimate the closure of schools in 185 countries, from which it is inferred that 89% of students did not attend classes in the first semester of 2020 due to the pandemic [6].

In the case of Chile, the Ministry of Education reported that it has only been able to cover 60% of learning in mathematics [7]. During the pandemic, the Latin American research agenda has highlighted the scarce evidence of how educators in virtual environments favor the learning of mathematics, alluding not only to digital platforms that exist to teach and learn mathematics but also to how such strategies can work technically and enable effective learning. [8]. As a result, the suggestions made by Papert [9] at the beginning of the 21st century on the need to provide valuable heuristics to design virtual learning have now become indispensable. The COVID-19 pandemic has posed multiple challenges for mathematics educators, such as the case of Khirwadkar [10], who has proposed teaching strategies that favor the participation of parents and students, using easily manipulated household items to understand abstract mathematical concepts.

It is known that many educators in 2020 were required to teach math courses online, with little training in how to teach blended or online formats [11]. It is of great importance to be able to understand from the perspective of educators the coaptation of new teaching strategies that have allowed them to respond to the challenges of online and distance mathematics education [12]. Providing information about the didactic strategies used by educators from different parts of the world regarding how they implemented their mathematics classes during the pandemic could be of great importance to the discussion about the implications of online education in the field of mathematics education and practice [13]. Current evidence on online mathematics pedagogical strategies has shown that social and research-based learning can guide mathematics education with different ways of taking advantage of technology and improving the online classroom environment [14].

In Chile, the curricular bases of mathematics education in primary school suggest the Concrete-Pictorial-Symbolic Method (CPS) proposed by Bruner [15] to promote cognitive development through the manipulation of concrete objects and pictorial and symbolic representations. However, it is unknown how elementary educators implemented their math classes during the pandemic. A relevant precedent is that the use of didactic strategies in Chilean classrooms is centered on the teacher and focused more on the exercise of procedures than on critical, creative, and metacognitive thinking skills [16]. Unlike existing evidence on this subject that describes the quality of online educational processes, the use of technological platforms, and new ways of learning [17], our study highlights the importance of how the teaching of mathematics has been carried out during the pandemic in urban and rural contexts. For this, we present the types of strategies used by primary educators who deployed different ways of teaching content. Therefore, this study could be very useful for researchers in the field of mathematics education, municipal educational corporations, and practicing educators who might use such platforms during periods of rising contagion in their educational organizations. This is due to the need to promote pedagogical reflection on more effective strategies according to different characteristics, among which the context, the use of educational resources, and years of experience predominate.

Based on the above, in this study, we address the following research question: What types of teaching strategies and educational resources do primary educators use in the subject of mathematics during their virtual classes for rural and urban schools in Chile? In relation to this question, we quantitatively describe the use of various didactic strategies in mathematics and the use of educational resources according to the respective contexts of the primary educators belonging to the sample. Based on the above, this research seeks to

respond to the problems detected regarding scarce existing knowledge about the types of didactic strategies in mathematics used by primary school teachers during online classes.

The research objectives are as follows:

1.  To socioculturally characterize the sample of primary educators who teach mathematics online.
2.  To identify the types of teaching strategies, educational resources, and educational contexts.
3.  To compare the types of didactic strategies that primary educators used in mathematics teaching according to the types of educational resources they used and the educational contexts in which they work.

## 2. Theoretical Framework

### 2.1. Online Didactic Strategies

The didactic strategies provided by educators comprise ways of encouraging students to learn content, develop skills, or develop an attitude toward problem-solving [18]. The design of teaching strategies in primary education requires placing students in scenarios that are attuned to their performance environments to provide them with opportunities that reveal the key characteristics of such situations [19]. Often the role of a mathematics educator is to encourage his/her students to follow a certain path and to respond sparingly to the questions or interests of the students. The new scenario of online education can influence the ways in which the teacher encourages his/her students to learn, either positively or negatively. This is because the teacher is the one who influences learning, so researching new teaching strategies is essential when designing a class [20].

During the course of the development of mathematics education, some ideas that worked well in a previous context may continue to be supportive in a new context, whereas others may indicate obstacles [21]. In this new framework of online education, many educators have had to reprogram and adopt new teaching strategies to encourage their students' learning. Educators and students from urban and rural areas throughout Uruguay had to transform their homes into classrooms, a situation that caused changes in traditional educational resources and teaching strategies [22].

With the arrival of the COVID-19 pandemic, educators have had to deploy new ways of teaching through online platforms such as Zoom, Meet, and Teams, among others [23]. The advantage that distance education offers is that it allows access for students and teachers from anywhere and at any time as long as there is an internet connection and a device [24]. Its implementation requires the incorporation of new technologies that take into account the different platforms to carry out the pedagogical processes through teaching strategies [25]. Kuntze [26] states that so-called "bottom-up" or emergent strategies are critical when teachers are encouraged to introduce alternative instructional practices. The digital literacy of educators and students is a predominant factor for all students to be able to learn in these new contexts [27].

### 2.2. Educational Resources of Online Education

Online education gives us the opportunity to extend learning outside the classroom and traditional teaching. In the framework of online education, the use of mobile devices (such as laptops, smartphones, and tablets) generates positive emotions in students toward learning mathematics [28]. During the pandemic, telephone and instant messaging applications such as WhatsApp, Telegram, and WeChat have been useful tools for teaching and learning mathematics, especially in rural contexts [29].

Ruthven [30] argues that new virtual technologies do not simply replicate the functionality of old resources with increasing efficiency, but also enable qualitatively different and unique forms of interaction. Several educators reported the benefits of Google Classroom for reflective and creative planning that engaged students with online resources [13]. These new forms of interaction depend not only on technology but also on educational resources such as learning guides, concrete materials, and domestic resources. This is how educational resources can be understood as "elastic curricular materials" that range from

sheets to digital resources in electronic formats that adopt the sequences of the curricular contents [31]. Through the COVID-19 pandemic, many teachers created their own white-boards by pasting papers on the walls of their homes to teach, whereas others recorded themselves on their phones so that their students could listen to them when they had an Internet connection [32].

The literature on didactic strategies in mathematics education is extensive; however, a general classification must consider the epistemological assumptions of the theories of cognition and learning to which they are ascribed [33]. In relation to this, we can distinguish traditional strategies linked to forms of teaching framed by cognitivism based on the automation of procedures, use of formalisms, heuristics, and inductive strategies. On the other hand, emerging strategies are ascribed to post-cognitivist approaches, such as embodied cognition, enactivism, and ecological psychology. Enactivism and ecological psychology assume that cognition is situated, embodied, and anti-representationalist [34]. Some emergent or enactive problem-solving strategies involve the explorations of students guided by their own understanding, creation, and meaning of these situations and tasks [35]. From this paradigm, emerging strategies in mathematics education can be classified into metaphors and analogies, and ostensive, inventive, and modeling practices.

## 3. Materials and Methods

### 3.1. Research Context

This research is framed in a post-positivist paradigm, based on a quantitative approach with a non-experimental descriptive design. This study was conducted in the context of the COVID-19 pandemic in primary schools in the Coquimbo Region in Chile. Following the closure of schools, the Chilean Ministry of Education decided in the context of online education to implement curricular prioritization as a support tool for schools that allowed them to select the minimum and essential learning objectives for each subject. This initiative is considered essential to better face and minimize the consequences of the pandemic on educational processes such as the scarce digital and technological training of educators and the connectivity problems of students in rural sectors [7]. Mathematics education in Chile is based on three pillars according to the curricular bases: skills, thematic axes, and attitudes [7]. The suggested skills for achieving mathematical learning are to model, represent, solve problems, communicate, and argue. The thematic axes correspond to the contents of numbers and operations, geometry and measurement, patterns and algebra, and data and probabilities. In relation to attitudes, a positive attitude, methodical work, creativity, curiosity, effort, and perseverance are favored.

To carry out the development of these skills and attitudes in each thematic axis, the curricular bases suggest applying the CPS method [36], which points out the importance of action and mediation for cognitive development. According to [37], Bruner distinguishes three stages of representation: "(1) an enactive or concrete form, in which students develop mathematical concepts by physically manipulating concrete targets; (2) a pictorial form, in which they learn to represent a mathematical concept in graphical or pictorial form, and (3) a symbolic form, in which they learn to represent a concept with an abstract model or symbols" (p. 2). Based on these ideas, it is suggested that the teaching of mathematics for effective and meaningful student learning should incorporate these three stages [38,39]. Primary educators in Chile conducted their classes using virtual platforms such as Meet, Teams, and Zoom.

### 3.2. Procedure and Sample

In this context, it was considered crucial to investigate the ways in which primary mathematics teachers implemented their classes, taking into account the sociocultural context, educational resources, level of technological literacy, and use of teaching strategies. The findings are expected to have implications on educational policy and planning to support online teaching and learning. Data was collected through an online questionnaire in September 2021; ethical issues were considered in accordance with the new General Data

Protection Regulation (GDPR), with participation in the survey being voluntary. Teachers were informed that the questionnaire was anonymous and that the data collected would be used for research purposes only. The questionnaire was delivered openly via email to teachers who were interested in participating in the study.

In this questionnaire, the closed questions referred to the sociodemographic characteristics of the sample of teachers, and the open-ended questions referred to educational interaction processes such as (1) the educational resources used and (2) the types of strategies. In the case of the open-ended questions, teachers' detailed descriptions of the types of resources and strategies used during online mathematics education were explored. Subsequently, we classify the descriptions of the strategies and quantify them according to use.

The sample consisted of 105 participants. All the teachers came from schools with private, municipal, or foundation-based types of administrative dependency. Demographics were distributed by age in ranges from 20 to 29, 30 to 39, 40 to 49, 50 to 59, and more than 60 years. A total of 83% of the participants worked in urban areas and 18% in rural areas. In addition, they were segmented into years of service in ranges between 1 and 40 years.

*3.3. Data Analysis*

For the analysis of the data, an online questionnaire was implemented, which consisted of 11 closed questions with alternatives, which was sent to the teachers by email, where the reason for the investigation was explained and what the questionnaire consisted of. In addition to pointing out that their answers were anonymous and confidential in order to learn about their experiences of the ways in which primary educators implemented their mathematics classes for urban and rural schools during the COVID 19 pandemic, the following data were used: SPSS and Cytoscape programs. The results were generated in a database through Google Forms, where the data was processed in two stages.

In the first stage, the results were quantitatively analyzed through descriptive and inferential statistics using the SPSS software. In particular, we used the CHI2 technique because it allowed us to relate variables with different levels of measurement.

In the second stage, to visualize complex networks, we used an open-source software platform, Cytoscape [38], which was created for integrating, visualizing, and analyzing measurement data in the context of networks. We combined quantitative and qualitative information, so this paper presents a scenario of how we can create a network according to expression data from Google Forms.

## 4. Results

*Descriptive Measures about the Sociocultural Characterization of Primary Mathematics Educators Who Teach Online*

The sample of participants consisted of a non-probabilistic sample, which corresponds to 105 mathematics teachers from schools belonging to different administrative units (state = 55, private = 14, and foundation = 36). The method used during the COVID-19 confinement in 2020 involved a total of 18 items that are presented in the results section. The items were evaluated through nine parameters: educational context, job experience, mathematical mention, curricular prioritization, digital literacy, duration of classes, use of CPS method, type of strategy, and substrategy.

The pilot sample consisted of 105 participants. All the teachers were from schools with a private, municipal, or foundation administrative dependency. Demographics were distributed by age in ranges from 20 to 29, 30 to 39, 40 to 49, 50 to 59, and more than 60 years. A total of 83% of the participants worked in urban areas and 18% in rural areas. In addition, they were segmented into job experience in ranges between 1 and 40 years. Below we present Table 1.

**Table 1.** Characteristics of the sample (*n* = 105 educators).

|  | Range | Woman | Men | Private | State | Foundation |
|---|---|---|---|---|---|---|
| Age | 20–29 | 8.7% | 8.3% | 21.4% | 5.5% | 8.3% |
|  | 30–39 | 56.5% | 33.3% | 35.7% | 45.5% | 58.3% |
|  | 40–49 | 26.1% | 38.9% | 28.6% | 32.7% | 27.8% |
|  | 50–59 | 8.7% | 13.9% | 14.3% | 12.7% | 5.6% |
|  | 60–69 | 0.0% | 5.6% | 0.0% | 3.6% | 0.0% |
| Educational context | Rural | 11.6% | 19.4% | 14.3% | 20.0% | 5.6% |
|  | Urban | 88.4% | 80.6% | 85.7% | 80.0% | 94.4% |
| Job experience | 1–9 | 34.8% | 36.1% | 35.7% | 27.3% | 47.2% |
|  | 10–19 | 47.8% | 47.2% | 50.0% | 56.4% | 33.3% |
|  | 20–29 | 13.0% | 11.1% | 14.3% | 9.1% | 16.7% |
|  | 30–39 | 4.3% | 5.6% | 0.0% | 7.3% | 2.8% |
| Mathematical mention | Yes | 37.7% | 27.8% | 28.6% | 36.4% | 33.3% |
|  | No | 62.3% | 72.2% | 71.4% | 63.6% | 66.7% |
| Curricular prioritization | Yes | 94.2% | 94.4% | 78.6% | 98.2% | 94.4% |
|  | No | 1.4% | 2.8% | 7.1% | 0.0% | 2.8% |
|  | Sometimes | 4.3% | 2.8% | 14.3% | 1.8% | 2.8% |
| Digital literacy | Low level | 4.3% | 11.1% | 7.1% | 7.3% | 5.6% |
|  | Medium level | 69.6% | 72.2% | 57.1% | 70.9% | 75.0% |
|  | High level | 26.1% | 16.7% | 35.7% | 21.8% | 19.4% |
| Duration of classes | 10–20 min | 1.4% | 0.0% | 0.0% | 1.8% | 0.0% |
|  | 21–30 min | 8.7% | 11.1% | 14.3% | 9.1% | 8.3% |
|  | 31–40 min | 20.3% | 27.8% | 7.1% | 29.1% | 19.4% |
|  | 41–50 min | 30.4% | 36.1% | 14.3% | 30.9% | 41.7% |
| Use of CPS method | No | 36.2% | 50.0% | 35.7% | 40.0% | 44.4% |
|  | Yes | 63.8% | 50.0% | 64.3% | 60.0% | 55.6% |
| Type of strategy | N. A. | 2.9% | 5.6% | 7.1% | 3.6% | 2.8% |
|  | Traditional | 42.0% | 47.2% | 57.1% | 40.0% | 44.4% |
|  | Emerging | 55.1% | 47.2% | 35.7% | 56.4% | 52.8% |
| Substrategy | Heuristic | 1.5% | 8.8% | 7.7% | 1.9% | 5.7% |
|  | Inductive | 4.5% | 2.9% | 7.7% | 1.9% | 5.7% |
|  | Process automation | 30.3% | 32.4% | 38.5% | 26.9% | 34.3% |
|  | Formalization | 7.6% | 5.9% | 7.7% | 11.5% | 0.0% |
|  | Metaphorical | 25.8% | 26.5% | 30.8% | 26.9% | 22.9% |
|  | Modeling | 3.0% | 0.0% | 0.0% | 3.8% | 0.0% |
|  | Ostensive | 1.5% | 0.0% | 0.0% | 0.0% | 2.9% |
|  | Analog | 24.2% | 17.6% | 7.7% | 23.1% | 25.7% |
|  | Inventive | 1.5% | 5.9% | 0.0% | 3.8% | 2.9% |

It can be seen that 99% of teachers indicated that they use curricular prioritization. However, the ways to implement it vary significantly between them. From the results, we can also see that only 24% of participants have a high level of literacy in the use of virtual platforms; on the contrary, 74% indicate that they are at a medium level. Only 7% reported having a low level of knowledge.

It is important to highlight that 84% of participants indicated the use of metaphors as a strategy for the representation of the development of abstract concepts through daily and concrete experiences. In addition, 93% of teachers indicated the use of mathematical modeling as a strategy for the representation of reality through mathematical notions to solve real problems. A particularly interesting finding was related to the fact that had we observed the way in which mathematical modeling was implemented and carried out in the classroom, we would have seen vast differences in teachers' understanding of mathematical modeling. Below we present Table 2.

**Table 2.** Comparison of teaching strategies, job experience, and educational resources.

| Type of strategy | Job Experience Traditional | | | | Job Experience Emerging | | | |
|---|---|---|---|---|---|---|---|---|
| | 1–9 | 10–19 | 20–29 | 30–39 | 1–9 | 10–19 | 20–29 | 30–39 |
| Learning guides | 13.3% | 48.0% | 50.0% | 50.0% | 0.0% | 0.0% | 25.0% | 33.3% |
| Concrete material | 33.3% | 24.0% | 50.0% | 0.0% | 61.9% | 43.5% | 62.5% | 33.3% |
| Use of Software | 53.3% | 24.0% | 0.0% | 0.0% | 23.8% | 30.4% | 12.5% | 0.0% |
| Resources of the domestic environment | 0.0% | 0.0% | 0.0% | 0.0% | 9.5% | 4.3% | 0.0% | 33.3% |
| Audiovisual | 0.0% | 0.0% | 0.0% | 50.0% | 0.0% | 0.0% | 0.0% | 0.0% |

We used the software Cytoscape to visualize the complex networks in Figure 1. So, we can see that the traditional strategy most used by primary mathematics educators is process automation, although metaphorization and analogy are the emerging strategies most used by educators during online teaching. In the first case, the emphasis on resolution over understanding is reaffirmed, whereas in the second, understanding is favored over resolution. Below we present Figure 1.

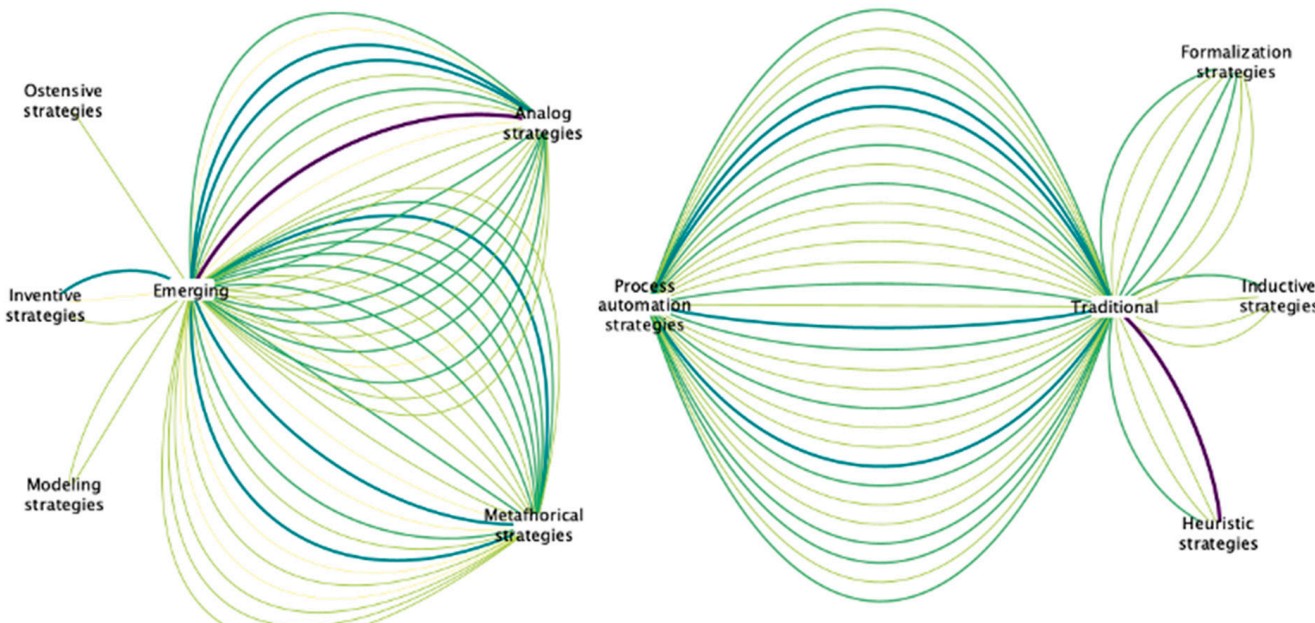

**Figure 1.** Complex networks of type of strategy and educational resources.

To answer our research question, we used the CHI2 technique, which allows us to compare the frequencies observed in the educational resource categories with the frequencies that would be obtained randomly. We can see in Table 3 that our independent variable is the type of strategy, and our dependent variable is the educational resource. In the null hypothesis (Ho) we find that there is no difference between the proportion of educational resources used by teachers who use traditional strategies and emergent strategies. So, if p(v) is less than or equal to 0.05, it is rejected and, therefore, we would assume that our independent and dependent variables are related. Below we present Table 3.

**Table 3.** Relationship between teaching strategies and type of educational resources.

| Educational Resources | Traditional | Emerging | Total |
|---|---|---|---|
| Learning guides | 17 | 3 | 20 |
| Concrete material | 13 | 29 | 42 |
| Use of software | 14 | 13 | 27 |
| Resources of the domestic environment | 0 | 4 | 4 |
| Audiovisual | 1 | 0 | 1 |
| TOTAL | 46 | 55 | 101 |
| Chi-Square Tests | | | |
| Pearson Chi-Square | Value | df | Asymptotic |
| Likelihood Ratio | 23,891 a | 5 | 0.000 |
| Linear-by-Linear Association | 27,197 | 5 | 0.000 |
| N of Valid Cases | 101 | 1 | 0.445 |

a—6 cells (50.0%) have expected count less than 5. The minimum expected count is, 46.

We can see in Table 4 that our independent variable is the educational context and our dependent variable is the type of substrategy. In the null hypothesis (Ho) we find that there is no difference between the proportion of substrategies used by teachers who work in urban and rural areas. Then, if p(v) is less than or equal to 0.05, it is rejected and, therefore, we would assume that our independent and dependent variables are related. Below we present Table 4.

**Table 4.** Relationship between substrategies and educational context.

| Substrategy | Rural | Urban | Total |
|---|---|---|---|
| Process automation | 7 | 13 | 20 |
| Metaphorical | 1 | 25 | 26 |
| TOTAL | 8 | 38 | 46 |
| Chi-Square Tests | Value | | |
| Pearson Chi-Square | 7637 a | df | Asymptotic |
| Likelihood Ratio | 8132 | 1 | 0.006 |
| Linear-by-Linear Association | 7471 | 1 | 0.004 |
| N of Valid Cases | 46 | 1 | 0.006 |

a—2 cells (50.0%) have expected count less than 5. The minimum expected count is 3.48.

We can observe in Table 2 the proportion of the different educational resources between traditional strategies (f = 46, 45.5%) and emerging strategies (f = 55, 54.5%). With $X^2$ = 23.891, df = 5, and $p < 0.000$, as the significance is less than 0.05, Ho is rejected, and so is the proportion of educational resources used by teachers who use traditional or emerging strategies.

In Table 4, we can see the proportion of the different substrategies between urban areas (f = 38, 82.6%) and rural areas (f = 8, 17.4%). With $X^2$ = 7.637, df = 1, $p < 0.006$, since the significance is less than 0.05, Ho is rejected, so the proportion of substrategies used by teachers who work in urban areas is different from teachers who work in urban areas.

## 5. Discussion and Conclusions

As a result of the COVID-19 pandemic, educational processes changed on all levels, forcing educators to quickly transform in-person classes into online classes. In just a few months, educators in much of the world launched a series of individualized organizational measures, focused on tutorials, made methodological and curricular adaptations, and monitored student progress. The pandemic not only transformed the contexts for implementing the curriculum by using online education platforms but also promoted teaching methodologies and strategies for which the curriculum was not designed.

The current online education framework has introduced new ways of teaching for developing learning skills in all disciplines. In this way, curricular adjustments and the use

of different pedagogical resources have made it possible to adapt to the challenges of online education. In the case of mathematics education, which is what concerns us, it represents an additional methodological challenge for primary school educators given the symbolic nature of the contents. Henceforth, we address the following question: What types of teaching strategies and educational resources do primary educators use in response to the pandemic in the subject of mathematics during their virtual classes for rural and urban schools in Chile? This study contributes to and expands the literature by offering new evidence about the teaching strategies and educational resources used by primary school educators during the COVID-19 pandemic when implementing their online mathematics classes in their respective contexts. For this, we first socioculturally characterized the sample of 105 teachers who answered an online questionnaire through Google Forms, in which we specifically explored the teaching strategies and educational resources, as well as the educational context in which they are used.

### 5.1. Sociocultural Characterization of Primary Educators Teaching Mathematics Online

Among the teaching strategies and types of learning to be promoted within mathematics education in the next 10 years are collaborative learning, critical mathematics education, dialogic teaching, modeling, personalized learning, and problem-based learning [39]. The findings of our study regarding the first research objective, which refers to the sociocultural characterization of the sample, show that 15.5% of the sample of educators belong to a rural educational context and 84.5% to an urban one. Likewise, there are higher percentages of female primary educators in the age group of 30–39 years, which is equivalent to 56.5%, whereas there are more male educators in the age group of 40–49 years with 38.9% of the total sample. Another important finding is that those who identified as having 10–19 years of work experience in schools were predominant, with 47.8% of women and 47.2% of men identifying with this level of experience. From this, we can conclude that the majority of the sample studied is primarily women under 40 years of age who have less than 20 years of experience. This means that they could be more open to change, given that they have become familiar with technology during their professional development [40].

In the characterization of the educational aspects of training, it can be seen that a majority of primary educators identify with a moderate level of digital literacy, with 69.6% corresponding to women and 72.2% to men. Only 26.1% of women and 16.7% of men identified themselves as having a high level of digital literacy. This is relevant to understanding how quickly primary school teachers adapted to online education. With a lower-than-average percentage of the level of digital literacy, it would have been difficult to effectively implement online classes. This is consistent with what the OECD [41] maintains, stating that the success of schools with little experience in using an online approach must be supported by a reliable infrastructure, supportive leadership, and trained and motivated teachers to achieve their goals.

### 5.2. Identification of Teaching Strategies, Educational Resources, and Type of Educational Context

Based on the second research objective referring to the identification of the didactic strategies deployed by primary educators as well as the resources they use and the educational context in which they work, it was shown that didactic strategies for online education can be classified as traditional and emerging. The traditional strategies found were the automation of procedures, the use of formalisms, and induction strategies. On the other hand, the emerging strategies that were reported focus on understanding the meaning of the task, such as metaphorization, analogy, and ostension and those that promote creativity and modeling. The use of these strategies is based on the fact that interaction in online contexts allows us to consider links between the processes of construction of meaning and participation that support learning [42].

The didactic strategies can be ascribed to conceptions of traditional learning ascribed to cognitivism since the idea is sustained that the central nucleus of cognition and learning in mathematics is processing, automation, and the use of formalisms together with the

induction process [43]. As suggested by [44], traditional didactic strategies place emphasis on the deductive and objectivation processes of mathematical knowledge, cracking the constitutive learner–environment mutualism. Instead, emerging strategies are usually ascribed to paradigms of embodied cognition and enactivism [45]. The foregoing becomes relevant to understanding whether online education, apart from suggesting new interaction spaces, transforms traditional practices of mathematics teaching, which are usually focused on solving problems and automating procedures [46]. Similarly, our findings allow us to respond positively to the concerns of [23] that the hasty adoption of new technologies leads to regression toward a pedagogy of knowledge transmission.

In relation to the emerging strategies, in this study, metaphorization strategies can be highlighted with usage by 25.8% of women and 26.5% of men. The use of analogy is also evident in 24.2% of women and 17.6% of men. Both teaching strategies are framed within cognitive semantics, but metaphors specifically emphasize the role of the body in cognition [47]. This is highly relevant for learning based on the experiences of each student who, through the involvement of perception and action, can see one thing in terms of another, generally from a more concrete to a more abstract one [48]. In the case of analogy, our findings can be aligned with the proposals of [49], that educators should use well-understood analogies that can explain correspondences and thus facilitate inferences between analogous objects using verbal and visuospatial support. This is consistent with the use of educational resources from students' homes to thematize the mathematical content used by educators through emerging didactic strategies.

### 5.3. Comparison of Didactic Strategies According to the Type of Educational Resource and Educational Context

Regarding the educational context and the use of educational resources, our findings report the use of learning guides, concrete material, software, domestic resources, and audiovisual resources. When comparing the relationship between the use of educational resources and the didactic strategies corresponding to the third research objective, it is found that during the pandemic, teachers who used traditional strategies preferred the educational resources of teaching and learning guides and the use of the online education platform. In contrast, educators using emergent strategies preferred to spend more time using concrete materials and educational gamification software. Another relevant aspect is that 53.3% of educators who use traditional strategies and who have between 1 and 9 years of experience used digital teaching resources proposed by the Ministry of Education such as the Sumo First App and interactive simulations of the CPS method. From this, it can be inferred that educators who use emerging strategies with concrete materials and gamification software seek to highlight creative thinking for finding multiple solutions to mathematical situations by reconciling tacit perspectives of action with domestic objects and digital games that stimulate learning by discovery. However, educators who use traditional strategies prefer to reaffirm automation procedures in digital resources that promote the CPS method for convergent or linear learning.

In relation to the strategies used in urban and rural educational contexts, our findings show statistically significant differences. Metaphorization is an emerging strategy that is preferred by educators in urban contexts, whereas in rural contexts, the strategy of process automation predominates. However, only 23.8% of educators who have between 1 and 9 years of work experience and who use emerging strategies use only educational software. In the case of domestic resources, it can be seen that the majority of educators who use emerging strategies make use of these, unlike those who use traditional strategies. These findings are related to the idea that educators had to go beyond the curricula and reorient content to students' homes to maximize understanding [50]. Likewise, this idea of reinterpreting the home as a type of educational context revisits the idea of self-directed learning in which students focus on their situational experience for learning with concrete materials. This is characteristic of approaches based on the action of perception for learning such as embodied cognition and enactivism. On the other hand, the persistence of

traditional strategies saw an increase in the use of learning guides and videos to promote learning, which can be linked to cognitivism due to the passive emphasis that the student has on the task.

One of the important conclusions that we obtained from the study was that online education did not generate major changes in pedagogical performance. The use of gamification, digital resources, and educational software merely reproduced repeated strategies that did not impact practice, since traditional strategies preserved the forms of teaching that are used in face-to-face education. Likewise, the emerging strategies did not imply large incorporations of digital resources to convey the content. By way of implication, our findings can serve as suggestions and guidelines for schools or municipal corporations that wish to design their educational improvement plans within the framework of professional teacher development for primary school teachers. In this regard, training processes can be oriented toward the use of various didactic strategies in the subject of mathematics. For example, an emphasis on didactic strengthening in order to encourage the development of sensorimotor skills for the development of the exploration, inquiry, metaphorization, analogy, and creativity of students when working with concrete and pictorial materials, as well as in the use of software. To this can be added the strengthening of online pedagogical management, such as managing the virtual classroom, monitoring the learning process, and promoting metacognition through forums. This is consistent with the work of Bond [51], who suggested providing more funding for professional and team development, prioritizing equity, designing collaborative activities, and using a mix of synchronous and asynchronous technology. This is linked to the findings proposed by Nikolopoulou [52] that suggest the maintenance of good high-level teaching practices and the improvement of digital culture through combined teaching approaches.

## 6. Limitations and Future Research

This study was carried out during the months of August and September 2021 and has some methodological limitations. First, the instrument used was the online questionnaire that explored, among other things, the didactic strategies and educational resources that were identified by the primary educators on the form without triangulating with more ecological techniques such as the observation of online classes or partial/total recordings. Second, the sample size is moderate, so it is suggested that future research includes a larger sample and explores broader aspects of the use of teaching strategies such as the type of platform, effective times for classes, and types of strategies used for individual and collaborative student work. Future research could incorporate longitudinal research designs to understand the transition from online education to hybrid and face-to-face education. It is suggested that subsequent studies incorporate naturalistic methods that, through qualitative research designs, allow for a deeper understanding of the perspectives of teachers.

**Author Contributions:** Conceptualization, R.V., C.M. and C.A.; methodology, S.R. and R.V.; software, S.R.; validation, R.V., C.M., S.R. and C.A.; formal analysis, S.R.; investigation, R.V. and C.M.; resources, R.V.; data curation, S.R., R.V. and C.A.; writing—original draft preparation, R.V., S.R., C.M. and C.A.; writing—review and editing, R.V. and C.A.; visualization, S.R.; supervision, R.V.; project administration, R.V., S.R., C.M. and C.A.; funding acquisition, R.V. All authors have read and agreed to the published version of the manuscript.

**Funding:** This research was partially funded from INNOVA STEAM Lab. The content is the content is solely the responsibility of the authors.

**Institutional Review Board Statement:** He study was conducted in accordance with the Declaration of Helsinki, and approved by the research Ethics Committee of the Faculty of Education, Universidad Santo Tomás, La Serena, Chile.

**Informed Consent Statement:** Informed consent was obtained from all subjects involved in the study. CODE: 11.320.012, (Approved 6 June 2021).

**Data Availability Statement:** Not applicable.

**Acknowledgments:** We thank Viviana Rivera Barahona, coordinator of the pedagogical technical unit of the Gabriel González Videla Educational Corporation, for helping to disseminate the survey in the educational organizations of La Serena. All individuals included in this section have consented to the acknowledgment.

**Conflicts of Interest:** The authors declare no conflict of interest.

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
