# Peer review of "Online Mathematics Education during the COVID-19 Pandemic: Didactic Strategies, Educational Resources, and Educational Contexts"

_education, doi:10.3390/educsci12070492_

Round 1

Reviewer 1 Report

The article concerns the topic of teaching in the COVID era. In the article in the introduction, instead of describing what the authors did in the work, the authors recommend focusing on specifying research goals. The weakness of the work is that the authors do not state what is innovative in their work. There are many such studies in relation to teaching in the COVID era. I recommend that you focus on this area and convince the reader that the article fills the gap. The authors list the research goals on page 4 of the work, so I recommend that you move them to an introduction that introduces the reader to the work. The well-described methodology lacks information about the respondents and how they are good both for research and in terms of advancement in conducting classes virtually. The authors list the research goals on page 4 of the work, so I recommend that you move them to an introduction that introduces the reader to the work. The number of respondents is quite small, which reduces the value of the job. At this point, it's important to say how many teachers work at the school so that the sample size can be understood. I like figure 4 - nice looking. The value of the work is influenced by statistical analysis, which is the strength of the work. In the discussion, I propose to refer specifically to the questions posed and to answer them. I recommend that the article be checked in terms of style and language.

Author Response

1. Paragraphs 4 and 5 of the introduction highlight the importance of the study and what is new about studies in the field of online education. Our focus is the didactic strategies that convey the content in diverse educators and contexts. We highlight the importance for the Chilean and South American context of the limited knowledge about didactic strategies in mathematics during online education, as well as the similarities with research findings in England.
2. The research objectives were removed from the introduction to invite the reader to follow the coherence of the study between the problem-research question-objectives-method-analysis technique.
3. Point 3.2 of the research procedure details aspects concerning the characterization of the study sample.
4. Significant improvements in relation to the answers to the research question are incorporated into the discussion. For this, we are guided by each specific objective

Reviewer 2 Report

The overall manuscript is neat and written concisely—with relevant information for existing literature. One aspect that you can focus on is correct punctuation. Some sentences are incorrect. There is no independent clause and, as a result, your sentences are incomplete. Another aspect that I needs improvement is consistency in reporting on aspects (e.g., references in the reference list, but also statistics). In addition, you need to clarify how you view your major concepts related to anything digital, virtual or online. Do you use the terms interchangeable? This has to be clarified (e.g., in a footnote).

Author Response

1. In section 3.2, the paragraph presenting the stages of the Analysis Plan was improved to clarify the sequence of the methodology.

2. In section 4.1, the first paragraph was added and the second was improved to specify the non-probabilistic sampling procedure (characteristics of the participants). In addition, the dimensions of the instrument that will be evaluated are presented in the Results.

3. Incomplete sentences were completed to facilitate understanding and cohesion of the arguments.

4. In the final paragraph of the discussion, the way in which we conceive the virtual or online of our study was incorporated.

Round 2

Reviewer 1 Report

Thank you for improving your paper. Good job. 

Author Response

Thank you very much for the suggestions and comments that have been very useful to improve the article.

In relation to your suggestions, we have added two empirical investigations cited in 11 and 14 respectively. Regarding the appointments, we consider that all are pertinent. Regarding the discussion, we consider that it is balanced according to the findings and the scope of the investigation.

Reviewer 2 Report

There is a period insert on page 2.84. Please check. I would structure the summary of parameters on page 6 with (a)..., (b)..., etc. Page 11.335 contains multiple ":.:". Please revise. In addition, there are still inconcistencies in the reference list (i.e., use of spaces between the initials, the use of a space before the addition "accessed on X"). Moreover, the hyphen and en dash use is still incorrect and inconsistent (between the pagenumbers you have to place an en dash instead of a hyphen). Also the capital letters use in the document titles is inconsistent (use capital letters for each essential word OR do not use that). Please make these revisions. 

Author Response

Thank you very much for the suggestions and comments that have been very useful to improve the article.

In relation to your suggestions, we have corrected all the suggestions made. The only suggestion that we couldn't correct was the blank page where we have nothing inserted and a space is presented by default. The summary of parameters according to the suggestions we did not understand much. We think it plays well as presented.